# Implementation strategies and economic considerations for point-of-care ultrasound in Low- and Middle-Income Countries: A scoping review

Grace W. Banda-Katha [1,2,3], Timothy Kachitosi[1,2,3], Henry C. Mwandumba[1,4], Victor Mwapasa[5], Lucky G. Ngwira[5], Benno Kreuels [1,3]*, Paul Rahden [3]

1 Kamuzu University of Health Sciences, School of Medicine, Blantyre, Malawi, 2 Queen Elizabeth Central Hospital, Department of Emergency Medicine, Blantyre, Malawi, 3 Bernhard Nocht Institute for Tropical Medicine, RG Neglected Diseases and Envenoming, Hamburg, Germany, 4 Malawi Liverpool Wellcome Trust, Blantyre, Malawi, 5 Kamuzu University of Health Sciences, School of Public Health, Blantyre, Malawi

☯ These authors contributed equally to this work.
\* kreuels@bnitm.de

## Abstract

Point-of-care ultrasound (POCUS) is increasingly adopted as a diagnostic tool in low- and middle-income countries due to its accessibility and comparatively low costs. This review aims to synthesise successful implementation strategies for POCUS and evaluate existing literature on its associated cost implications. We conducted a scoping review following recommendations of the Joanna Briggs Institute's manual for evidence synthesis and the PRISMA-Scoping review guidelines. The search included five databases (PubMed, Ovid Medline, Embase, Cochrane Library and CINAHL) identifying original research published before 30th of April 2025 addressing POCUS implementation or economic evaluation in low- and middle-income countries. The Consolidated Framework for Implementation Research and the associated Expert Recommendation for Implementation Change (CFIR-ERIC) matching tool were applied to assess implementation strategies which improve implementation processes. Six studies focusing on implementation strategies were identified, five of which were conducted in sub-Saharan Africa. An additional six economic evaluations were included, mainly from Africa and South Asia. Implementation studies were primarily small-scale, single-country interventions with infrequent use of needs assessments prior to implementation and limited long-term follow-up. Common barriers included limited infrastructure, equipment shortages and delays in delivering intervention services. The CFIR–ERIC tool identified supportive strategies such as thorough planning, close collaborations involving several stakeholders, expanded training efforts and infrastructure, and continuous re-evaluations of interventions. Economic evaluations were methodologically diverse, often indication-specific and

**Data availability statement:** Yes - all data are fully available without restriction; All relevant data are within the paper and its Supporting Information files. The raw data underlying the extraction process, are publicly available via the Open Science Framework (OSF) repository and can be freely accessed at: https://doi.org/10.17605/OSF.IO/BZCQA.

**Funding:** The conduction of the study was supported by a grant from the Else-Kröner-Fresenius Stiftung (grant no: 2022_EKHA.155), awarded to the senior author BK. The funders had no role in study design, data collection and analysis, decision to publish, or preparation of the manuscript.

**Competing interests:** The authors have declared that no competing interests exist.

focused on cost analysis. Evidence on POCUS implementation and economic evaluation in low- and middle-income countries was diverse. Strategies such as thorough planning, close and equitable partnerships, and continuous re-evaluation and adaptation of interventions were identified as supportive of successful implementation. While potential cost savings to health systems have been reported, future efforts should prioritise comprehensive evaluations of POCUS implementation programmes that incorporate patient-centred economic evaluations assessments.

## Introduction

Medical imaging is an integral component of modern healthcare, however, access to advanced imaging modalities remains limited in many low- and middle-income countries (LMICs) [1]. Given its comparatively low financial and technical requirements, ultrasound offers the most accessible imaging option for these settings [2]. However, comprehensive ultrasound examinations require a high level of training of personnel and are often conducted by specialized technicians (sonographers) or doctors (radiologists). These cadres of specialists remain scarce in many LMICs and access to high-quality ultrasound services is limited [3].

Point-of-care ultrasound (POCUS) offers the possibility of real-time bedside diagnostic assessments and procedural guidance and is performed by attending clinicians [4]. Rather than a comprehensive ultrasound assessment, POCUS concentrates on a pragmatic approach, focusing on specific questions that immediately influence patient management [4]. It has been shown to be timely and to improve patient management across multiple disciplines [5–7]. POCUS may therefore have great potential in LMICs [8]. Several dedicated POCUS protocols have been developed to address common clinical challenges in these settings, such as trauma, heart failure, liver cirrhosis, maternal and neonatal health conditions, tuberculosis and other infectious diseases [9–13]. However, these protocols have so far been implemented only sporadically, limiting their broader impact.

Recognising these benefits, there has been an increasing interest in POCUS and several programmes have aimed to integrate POCUS into clinical practice across a variety of countries, disciplines and indications, driven by different stakeholders with diverse aims and priorities across individual programmes [14]. However, such initiatives rarely report strategies or success rates and recommendations for implementation strategies are lacking.

To ensure long-term sustainability, standardised yet adaptable approaches to implementation would be valuable to optimise the use of available resources, particularly in LMICs. Additionally, identifying common challenges and key facilitators for integration would be highly beneficial.

The Consolidated Framework for Implementation Research (CFIR) provides a comprehensive approach to identify barriers and facilitators in complex implementation projects [15,16]. Unlike other frameworks, it explicitly addresses factors related to an innovation itself, outer and inner contextual setting, the characteristics of

individuals involved and the implementation process [15,16]. In addition, the CFIR-Expert Recommendations for Implementing Change (CFIR-ERIC) tool was developed to generate tailored implementation strategies based on identified barriers to improve implementation processes [17,18].

Although the literature on POCUS is expanding, reviews published to date focus on training or educational methods rather than comprehensive implementation strategies [14,19,20]. None have been conducted that consolidate implementation strategies and evaluate their economic implications in LMICs. This review therefore aims to (i) identify and describe strategies that have been used to implement POCUS in LMICs and how they have been evaluated, (ii) highlight challenges and sustainable approaches for integration of POCUS into routine clinical practice, and (iii) evaluate the evidence on the economic evaluation of these strategies to inform the development of future POCUS implementation strategies and facilitate evidence-based decision-making by stakeholders.

## Materials and methods

### Study design and search strategy

We conducted a scoping review following the recommendations of the manual for scoping reviews by the Joanna Briggs Institute (JBI) [21] and PRISMA guidelines (S1 Checklist) [22]. For this review, we aimed to include articles mapping implementation strategies for POCUS as well as economic evaluations of POCUS implementation efforts. These components inform the feasibility of programme implementation and the effective allocation of resources. Consequently, a systematic search was conducted in PubMed, Ovid Medline, Embase, Cochrane Library and CINAHL for original research articles published before 15 February 2024 and an updated search for new literature was performed on 30 April 2025. The initial search strategy was designed for PubMed including the synonyms for 'POCUS' AND 'developing countries' AND ('implementation' OR 'cost-benefit' OR 'cost-effectiveness') and was adapted for further databases. Synonyms for 'developing countries' included individual countries defined as LMICs by the World Bank as of 2025 [23]. The complete search strategy was published with the registered protocol and is accessible on Open Science Framework and in S1 Text [24].

### Eligibility criteria

Eligible studies consisted of original research involving human subjects in a health-facility environment in LMICs. No restrictions were applied regarding patient age or publication language.

Studies were excluded if they did not contain original data or the available text did not provide sufficient information (e.g. only conference abstract available). Additionally, studies in which ultrasound examinations were performed by radiologists or sonographers or other specialised ultrasound operators (e.g. cardiologists) were excluded, as this was considered expert ultrasound rather than a point-of-care approach by the clinician responsible for the patient.

### Screening Process

All studies identified through the systematic search were extracted from the respective databases and transferred to Rayyan, a platform for managing systematic reviews [25]. After removing duplicates, titles and abstracts were independently screened by two reviewers (selected from a team of three involving TK, GK and PR) blinded to each other's decisions. Disagreements were resolved by consensus with a third reviewer. Two reviewers (GK and PR) conducted the full text review independently and resolved disagreements by consensus.

### Data management

Data were extracted by single reviewers (GK and PR) using standardised Microsoft Forms. Extracted information included authors, affiliations, title, year of publication, objectives, study setting and location, study period, study design, sample size, reported barriers and facilitators of implementation and reported study outcomes. Results were summarised in Microsoft Excel, extracted data were aligned for consistency and reported descriptively.

 

Implementation reports were analysed using the CFIR-ERIC tool [26]. Barriers and facilitators of the implementation process were extracted from included studies. To obtain individual evidence-based strategies to improve implementation processes, extracted barriers were entered into the CFIR-ERIC matching tool. The CFIR-ERIC tool generates percentage scores reflecting the relative strength of specific expert recommendations, enabling a prioritised approach. As barriers were rarely reported, we additionally applied the tool in reverse by inputting acknowledged facilitators reported in the studies, allowing us to identify enablers of successful implementation. This approach was previously explored experimentally [27,28]. The tool was applied to each study individually, once for barriers and then repeated for facilitators. The ten most commonly recommended strategies and facilitators were summarised for this review.

Figures were created using the open-source software tools Draw.io [29] and MapChart.net [30].

## Quality assessment

Included studies were critically appraised to evaluate their methodological quality using tools developed by the JBI. For studies employing a before-and-after design that involved the implementation of a POCUS intervention, we applied the checklist for quasi-experimental studies [31] following the recommendations of the JBI Manual for Evidence Synthesis [32]. For studies focusing on economic evaluation, we used the JBI checklist for economic evaluations [33]. We amended the checklists to allow for a "partially fulfilled" option, to better reflect individual aspects of included studies. The full tools can be found in S2 Text. For each response, points were assigned if a question was answered with yes (2 points) or partially (1 point). Based on the assigned score, studies were initially categorised as either high (≥75% of possible points), moderate (≥50–74%) or low (<50%) quality independently by two reviewers (GK, PR) and final categorisation was made in discussion between authors.

## Results

The search identified 1,476 records, of which 1,253 remained after removal of duplicates. Following title and abstract screening, 1,213 studies were excluded resulting in 40 studies that qualified for full-text review. Of these, 28 were excluded as they did not meet the inclusion criteria (Fig 1). Studies focusing on training initiatives rather than the implementation of an intervention were excluded as training approaches for POCUS have been described elsewhere [14]. In total, 12 studies were included in the final analysis (Fig 1). All included studies were published between 2014 and 2024.

Implementation programmes were described in six publications [34–39], while another six publications reported on economic evaluations of ultrasound-related interventions [40–45]. Five of the implementation reports were from initiatives in African countries (Fig 2), specifically Ethiopia, Tanzania, Malawi, the Republic of Congo and Ghana [34,35,37–39], while one was conducted in South America (Peru) [36]. Economic evaluations were performed in five African countries (Rwanda, the Democratic Republic of Congo (DRC), Kenya, Zambia and Egypt) [40,41,43], three Asian countries (Pakistan, India and Saudi Arabia) [40–42,44,45] and Guatemala [40]. While five economic evaluations incorporated single countries, one study performed an economic evaluation across five countries, specifically DRC, Guatemala, Kenya, Pakistan and Zambia [40].

## Quality assessment

The methodological quality of the studies included in this review varied. Among the six implementation studies, two were of high methodological quality [37,38], three were of moderate quality [35,36,39], and one was of low quality [34] (S3 Text). Of the studies reporting an economic evaluation, three were rated as high quality [40,43,44] and three as moderate [41,42,45] (S3 Text).

## Studies on POCUS implementation

Of the six studies investigating implementation programmes (Table 1), four were designed as prospective observational studies conducted alongside ultrasound training courses [36–39]. Two studies employed a cross-sectional design [34,35].

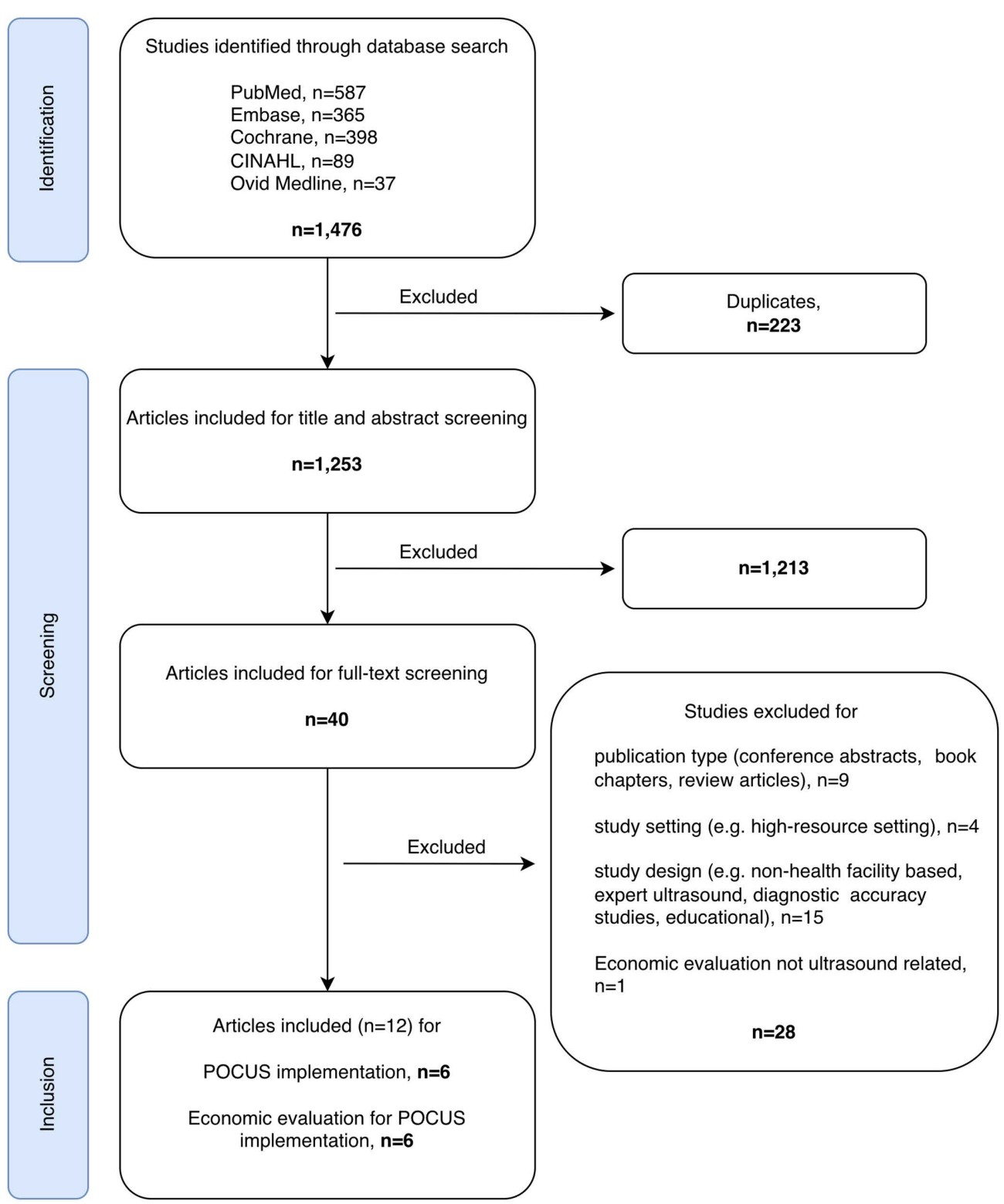

**Fig 1. PRISMA Flow Diagram: Selection Process for Studies on POCUS Implementation and Economic Evaluation in Low- and middle-income countries (created with draw.io).**

PLOS Global Public Health

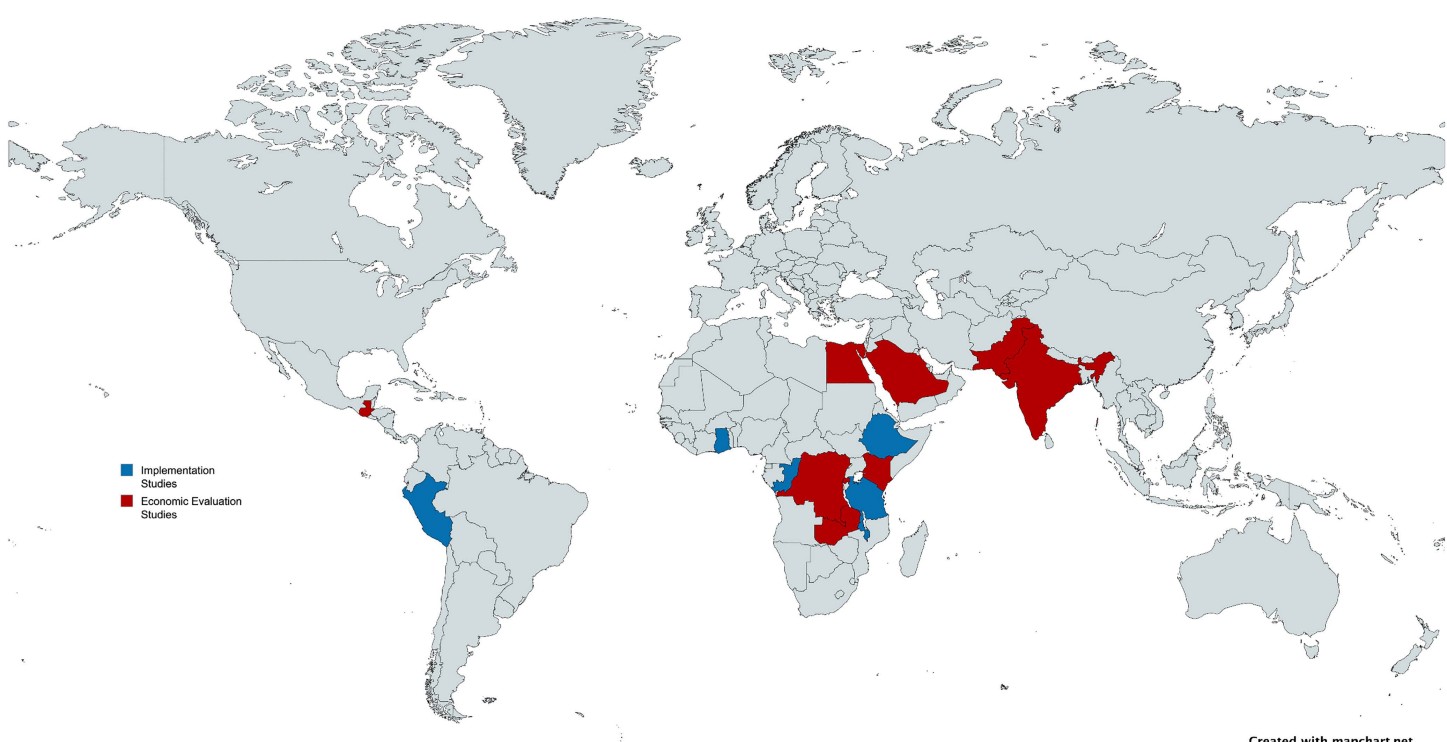

**Fig 2. Global distribution of focus area in studies included; blue=studies included for implementation programmes, red: studies included for economic evaluation.** The map was created using MapChart.net and was published under the Creative Commons Attribution 4.0 International License (CC BY 4.0) as granted by Minas Giannekas, owner and founder of MapChart (original copyright 2023). Map base layer: https://www.mapchart.net/world.html.

Implementation efforts targeted specific clinical departments, with four studies focusing on obstetrics/antenatal care particularly in primary and secondary care facilities [34,37–39] involving either midwives alone [38,39] or midwives and clinicians [34,37]. Two studies focused on emergency medicine physicians in tertiary care centres [35,36].

Needs assessments prior to POCUS implementation were conducted in only two studies [35,37]. While most programmes provided in-person hands-on training lasting two weeks for POCUS introduction [34,36–38], one study evaluated the impact of a one-year fellowship [36] and one programme conducted four training sessions over a two-years period [39]. This programme was embedded in a larger trial which utilised trained staff to assess gestational age during pregnancy [39].

To assess knowledge and skill transfer, three studies utilised written pre-/post-tests or objective structured clinical examinations (OSCE) of trained staff directly before and after the trainings [35,37–39]. Remote support and/or mentorship regarding image review for quality assurance were offered in four studies [35,37–39], while real-time support and feedback was provided in two studies [34,36].

To evaluate implementation success, all studies used the number of trained qualified staff as the primary outcome measure [34–39], while some studies reported on additionally trained instructors [35,36]. Skill acquisition and retention over time up to six months post-intervention were reported as successful implementation outcomes in two studies [37,38]. One study reported the number of scans, protected scanning time for fellows, change in patient care protocols and expansion of POCUS use in other departments as a result of POCUS implementation [36]. This also marked the only implementation strategy that incorporated the development of research skills (e.g., preparation of scientific publications) as a component of capacity building [36]. One study reported on the adaptation of a POCUS course curriculum based on a needs

**PLOS Global Public Health**

Table 1. Summary of studies focusing on POCUS implementation programmes in LMICs.

| Author, year | Country, level of care | Study design | Participating target group/POCUS application | Study aim | Needs assessment | Implementation method | Pre-/Post tests | OSCE | Image Review/Quality assurance | Remote support | Follow-up assessments | Completion/competency rate | Trained local instructors | Skill acquisition and retention | Feasibility aspects | Further |
|---|---|---|---|---|---|---|---|---|---|---|---|---|---|---|---|---|
| | | | | | | | | | | | | | | | | |
| Aeberlie et al., 2018 [34] | Republic of Congo, primary care | Cross-sectional study | • One midwife, one nurse and one clinician • Obstetrical care | • Evaluating the feasibility of real-time tele-training for obstetrical ultrasonography | | • Initial two weeks in-person training on POCUS • Support from international staff (Switzerland) | | | ✓ | ✓ | ✓ | ✓ | | | ✓ | |
| Aspler et al., 2022 [35] | Ethiopia, tertiary care | Cross-sectional survey | • 24 emergency medicine physicians and trainees • POCUS for acute and emergency care | • Identifying POCUS protocols with the highest clinical benefit • Creating a self-sustaining POCUS training programme | ✓ | • Identification of three priority POCUS applications for a training curriculum among POCUS trainees • Instructor training to enhance local capacity • Support from international staff (Canada) | ✓ | | ✓ | ✓ | ✓ | ✓ | ✓ | | | • Established core protocols include FAST, LUS and IVC-scans |
| Boamah et al., 2014 [39] | Ghana, secondary care | Prospective longitudinal study | • 15 midwives • Ante-partum obstetrical care | • Introducing ultrasound guided gestational age assessments | | • Intermittent ultrasound training over a 2-years period for basic and advanced assessments of pregnancy and fetal status • Establishment of standard operating procedures for estimating gestational age • Integration of ultrasound skill into routine practice • Support from international staff (US) | | | ✓ | ✓ | ✓ | ✓ | | ✓ | ✓ | |

*(Continued)*

# Table 1. (Continued)

| Author, year | Country, level of care | Study design | Participating target group/POCUS application | Study aim | Needs assessment | Implementation method | Evaluation/Follow-up | | | | | Reported outcomes | | | | |
|---|---|---|---|---|---|---|---|---|---|---|---|---|---|---|---|---|
| | | | | | | | Pre-/Post tests | OSCE | Image Review/Quality assurance | Remote support | Follow-up assessments | Completion-/competency rate | Trained local instructors | Skill acquisition and retention | Feasibility aspects | Further |
| Dreyfuss et al., 2020 [36] | Peru, tertiary care | Prospective cohort study | • Three Emergency medicine physicians • Advanced POCUS application | • Assessing a year-long ultrasound fellowship to enhance education in low-resource settings | | • Initial 2-weeks in-person training on POCUS • Clinical practice with protected time for scanning for one year • Support by US trainers in-person and remotely via real-time scanning • One-month rotation to the US • Support from international staff (US) | | | ✓ | ✓ | ✓ | ✓ | ✓ | ✓ | | • Acquisition of scientific competencies • Further adoption of POCUS in several departments • Regular rotations for POCUS trainees nationwide |
| Hall et al., 2021 [37] | Tanzania, primary to secondary care | Prospective longitudinal study | • 8 midwives, three clinical officers, two physicians • Antepartum obstetrical care | • Assessing the implementation of ultrasound use | ✓ | • Tailored POCUS training programme for 2-weeks • Six months supervised scanning during routine practice • Image review for technical quality and interpretation • Support from international staff (US) | ✓ | ✓ | ✓ | ✓ | ✓ | ✓ | | ✓ | | |
| Viner et al., 2022 [38] | Malawi, primary to secondary care | Prospective cohort study | • 29 midwives • Antepartum obstetrical care | • Evaluating a remote training programme for instructors with ultrasound expertise • Enabling subsequent teaching of midwives | | • Virtual train-the-trainers programme • POCUS training programme for 2-weeks • Group discussions to evaluate potential challenges • Integration of POCUS skill into routine practice • Support from international staff (UK) | ✓ | ✓ | ✓ | ✓ | ✓ | ✓ | | ✓ | ✓ | |

FAST: Focused Assessment with Sonography in Trauma, IVC: inferior Vena cava, LUS: Lung ultrasound, OSCE: Objective structured clinical examination, POCUS: Point-of-care ultrasound, US: Unites States, UK: United Kingdom

assessment of trained staff in Ethiopia [35]. Technical feasibility aspects like internet connectivity in real-time teaching were assessed in two studies [34,38]. All studies received support from international staff based in the United States of America (USA) [36,37,39], United Kingdom (UK) [38], Canada [35] or Switzerland [34].

Concerning enabling factors for successful implementation, all studies mentioned adequate study planning and continuous reflection and evaluation of the implementation process (Table 2). Additionally, supporting key individuals who led the intervention (champions) and the involvement of key stakeholders outside of the implementing facility to support the process (external change agents) were frequently mentioned. A well-established network of organisations and stakeholders (Cosmopolitanism, Networks & Communications) were described as beneficial.

Overestimating available resources (e.g., internet connectivity, available ultrasound equipment and consumables), delays in execution of the intervention (e.g., due to strikes, COVID-19 pandemic), as well as identification of the intervention being too complex were the most commonly mentioned barriers (Table 2).

When we entered barriers into the CFIR-ERIC tool [26], the tool suggested strategies to overcome obstacles in the implementation process including expanding training efforts, improving funding and physical infrastructure, and adapting and reassessing the intervention (Fig 3). While these strategies were ranked highly when barriers were entered, additional strategies were identified as effective both in response to reported barriers and as enabling strategies when facilitators were input into the tool. These included the development of a comprehensive implementation plan including the assessment of barriers and facilitators and the establishment of continuous knowledge exchange. The input of facilitators into the CFIR-ERIC tool additionally identified close professional networks, interactive discussions and feedback sessions to continuously review the implementation progress as well as promotion of individual implementation leaders as key strategies supporting POCUS implementation.

## Economic evaluation of POCUS implementation

The six studies investigating economic implications of implementing POCUS for specific indications in LMICs were summarised in Table 3 [40–45]. These studies varied widely in the type of economic evaluation performed, their objectives and the applied methodologies.

One study from Pakistan performed a cost-utility analysis using a decision tree model to compare different screening strategies for developmental dysplasia of the hip, reporting disability-adjusted life years (DALYs) averted [44]. One study conducted a cost-effectiveness analysis comparing ultrasound to endoscopy for screening for oesophageal varices in Egypt and Saudi Arabia and reported cost savings per case detected [41]. A cost-minimisation analysis from India compared the expenses of ultrasound-guided plexus blocks to general anaesthesia in upper limb surgery [45]. The remaining three studies primarily assessed costs for the implementation of ultrasound programmes without measuring health outcomes and benefits [40,42,43]. One study used time-driven activity-based costing to estimate the health system costs of a breast cancer early diagnosis programme in Rwanda [43]. While this study did not assess patient outcomes or savings, different models were described indicating cost-saving potential through decentralised ultrasound. The second study estimated costs of antenatal ultrasound services across five countries [40] which was conducted in parallel to a cluster-randomised trial assessing its implementation at health centre level [46]. The third study estimated patient incurred costs of attending a specialised cardiac clinic. Authors assessed the proportion of referrals that could have been avoided and the proportion of severe cases that might have been missed if POCUS had been available on district level. In this cost analysis, authors modelled travel cost savings from implementing POCUS for cardiac triage in rural India but did not quantify health outcomes [42].

Five of the six studies adopted a health system perspective, focussing on costs incurred and/or saved by providers or public health systems [40,41,43–45]. One of these included patient-centred outcome evaluating DALYs averted [44]. Only one study focused on indirect costs, specifically travel expenses and lost income for patients [42]. Direct medical costs

**Table 2. Characteristics of implementation studies identified as Facilitator (F) and/or Barriers (B) categorised according to the Consolidation Framework for Implementation Research (CFIR).**

| Author, year | Planning | Reflecting & Evaluating | External change agents | Champions | Key Stakeholders | Cosmopolitanism | Networks & Communications | Access to knowledge and information | Formally appointed internal implementation leaders | Executing | Available Resources | Opinion leaders | Compatibility | Evidence Strength & Quality | Trialability | Learning Climate | Leadership Engagement | Complexity | External Policy & Incentives |
|---|---|---|---|---|---|---|---|---|---|---|---|---|---|---|---|---|---|---|---|
| Aeberli, 2018 | F | F | F | | | F | | | | B | B | | | | | | | B | |
| Aspler, 2022 | F | F | F | F | F | | F | F | B | F F | B | | | | | | | | |
| Boamah, 2014 | F | F | F | F | F | F | | F | | F | F | | F | | | | | B | |
| Dreyfuss, 2020 | F | F | F | F | F | F | F | F | F | | F | | | | | | | | |
| Hall, 2021 | F | F | F | F | F | F | F | | F | B | B | F | | | | | | B | F |
| Viner, 2022 | F | F | | F | F | | F | | | B | | F | | F | F | F | F | | F |

Facilitator (F), Barriers (B), or both (B/F)

**ERIC strategies derived from barriers**

- Conduct ongoing training (4)
- Access new funding (3)
- Change physical structure and equipment (3)
- Conduct cyclical small tests of changes (3)
- Promote adaptability (3)
- Purposely reexamine the implementation (3)

**ERIC strategies derived from both**

- Assess for readiness and identify barriers and facilitators (6/4)
- Develop a formal implementation blueprint (6/5)
- Capture and share local knowledge (5/3)
- Create a learning collaborative (4/4)

**ERIC strategies derived from facilitators**

- Build a coalition (6)
- Conduct local consensus discussion (6)
- Identify and prepare champions (6)
- Facilitation (5)
- Inform local opinion leaders (4)
- Organise implementation team meetings (2)

**Fig 3. Consolidated Framework for Implementation Research- Expert Recommendations for Implementing Change (CFIR-ERIC) strategies [17,18] derived from facilitators and barriers identified across implementation studies (frequency in brackets; for ERIC strategies supported by both, the frequency for facilitators is mentioned first).**

like personnel, consumables and diagnostics were included in all [40–45] and two studies included start-up costs such as ultrasound equipment purchases and training [43,45].

Three studies concluded that the implementation of ultrasound programmes was cost-effective and cost-saving by reducing the need for expensive procedures [41,43,45]. Pigeolet et al. concluded that implementing targeted ultrasound in combination with clinical examination was cost-effective for identifying hip dysplasia in children though not necessarily cost saving [44]. In contrast, one study concluded that the introduction of routine ultrasound in antenatal care would not be cost-effective for LMICs given the limited evidence supporting its clinical benefits [40]. Kimura et al. demonstrated that POCUS could reduce unnecessary referrals and travel costs for patients but highlighted the trade-off of potentially missing cases requiring immediate care [42].

**Table 3. Summary of POCUS studies focusing on economic evaluations in LMICs.**

| Author, year | Country | Study Design | POCUS Application | Cost Data Source | Population or sample used for cost estimation | Costs assessed | | | | | | Outcomes and findings |
|---|---|---|---|---|---|---|---|---|---|---|---|---|
| | | | | | | Set-up costs | Personnel | Consumables | Clinical care | Operational costs | Patient costs | |
| Bresnahan et al., 2021 [40] | DRC, Guatemala, Kenya, Pakistan, Zambia | Cost analysis of routine antenatal screening ultrasound | Antenatal care | Interviews with hospital administrators and reimbursement schedules | • 24,008 participants in intervention (ultrasound) arm <br> • 22,896 participants in control arm (no ultrasound) | | ✓ | ✓ | ✓ | ✓ | | • Antenatal care ultrasound screening increased visit costs <br> • Highest observed cost difference in DRC, lowest in Guatemala and Pakistan |
| Elrazek et al., 2015 [41] | Egypt and Saudi Arabia | Cost-effectiveness analysis of ultrasound screening for oesophageal varices and cancer | Gastroenterology/ Oesophageal varices and cancer screening | Average procedural costs at the respective facility | • Based on 1,100 ultrasound investigations | | (✓)[1] | (✓)[1] | (✓)[1] | (✓)[1] | | • Ultrasound identified patients at higher risk of oesophageal varices <br> • Ultrasound as an initial screening tool lowered costs compared to direct endoscopy |
| Kimura et al., 2024 [42] | India | Cost analysis of POCUS to reduce travel costs to access cardiac care in rural areas of India | Cardiology | Self-reported travel costs and loss of income | • 219 interviewed patients | | | | | | ✓ | • POCUS avoided approximately half of referrals to the tertiary centre by identifying patients not requiring further investigation <br> • Some patients requiring immediate care were missed when POCUS was used in place of formal echocardiography |
| Nambaziira et al., 2022 [43] | Rwanda | Cross-sectional cost analysis of a breast cancer diagnosis programme | Breast cancer screening | Human resource records, hospital finance records, annual budget reports | • 817 individual patient evaluated with different clinical pathways | ✓ | ✓ | ✓ | ✓ | ✓ | | • Personnel and consumables drive costs <br> • Biopsy and pathology services were the most expensive services <br> • Ultrasound reduced the cost by avoiding unnecessary investigations |
| Pigeolet et al., 2024 [44] | Pakistan | Cost-utility analysis of screening strategies using a decision model | Paediatrics, developmental dysplasia of the hip screening ultrasound | Financial department of the Indus Hospital & Health Network based on regional health system data | • Modelling of 1,000–30,000 newborns based on local birth rate | | ✓ | ✓ | ✓ | ✓ | | • Clinical examination and targeted ultrasound showed the biggest gain in DALYs averted per cost <br> • Comprehensive ultrasound averts more DALYs but requires a larger financial cost |

*(Continued)*

Table 3. (Continued)

| Author, year | Country | Study Design | POCUS Application | Cost Data Source | Population or sample used for cost estimation | Costs assessed | | | | | | Outcomes and findings |
|---|---|---|---|---|---|---|---|---|---|---|---|---|
| Ponde et al., 2016 [45] | India | Cross-sectional cost minimisation analysis of ultrasound guided brachial plexus blocks | Anaesthesia | Average costs at the respective facility | • 90 patients receiving plexus block compared to estimates of general surgery | ✓ | | ✓ | | ✓ | | • Costs of ultrasound guided brachial plexus block was lower than general anaesthesia mainly driven by extended time of the surgery<br>• Set-up costs could be recovered within 3 years |

DALY: Disability-adjusted life years; DRC: Democratic Republic of the Congo; POCUS: Point-of-Care Ultrasound

[1] Authors mention overall costs for gastric scope and ultrasound exams, however, further information is missing

## Discussion

This review identified six studies on POCUS implementation and six that performed economic evaluations of POCUS in LMICs. Implementation studies were project-specific interventions with a maximum of 30 participants, had a focus on specific clinical indications, were mostly restricted to single institutions and focussed on short-term training outcomes. Economic evaluations were diverse and focused on specific indications. They were primarily stand-alone assessments rather than evaluations embedded within implementation programmes, often evaluating health system savings while lacking patient-centred perspectives.

Our review identified gaps across implementation studies in study designs and assessments of sustainability among POCUS projects. Implementation frameworks, such as the Knowledge-to-Action process (KTA) were designed to support standardised and successful introductions of interventions. The KTA recommends performing needs assessments and evaluations of potential challenges prior to implementation to improve long term sustainability [47], which was rarely reported in the identified studies [35,37] and none of the studies used formal frameworks to inform the design of their initiatives. The frequently adopted before-and-after assessment of participants skills [35,37,38] as indicator for successful implementation provided little evidence of sustained uptake or long-term integration into clinical practice. Case-based evaluations assessing POCUS scans performed in daily practice, together with long-term assessments of how POCUS influences clinical decision-making, as recently demonstrated in a large-scale Kenyan initiative, published in August 2025, may improve this key point of implementation processes [47,48].

To optimize standardised recommendations for implementation processes, we applied the CFIR-ERIC matching tool in both its intended manner and in reverse to strengthen the output for expert recommendations supporting implementation strategies [17,18]. During the continuous development of CFIR and the ERIC tool, framework developers and previous authors acknowledged challenges regarding the adaptability and use in routine practice outside research settings as frameworks were often considered too complex to apply effectively [16]. Previous studies have applied the tool in a similar reverse manner, using facilitators as additional inputs, and reported substantial overlap with standard barrier outputs as well as additional insights due to increased data availability [27,28]. Consistent with their findings, our results suggest that the reverse application can strengthen the interpretation of implementation recommendations. However, strategies identified solely through the reverse application of the tool should be interpreted with caution given this experimental approach.

Key recommendations for implementation strategies identified included thorough planning and adaptation of needs-based training programmes, while strengthening partnerships. A previous review focusing on POCUS training programmes outlined challenges in resources allocation, maintenance or programmes and availability of local trainer capacity

[14]. These findings on a project-based level in addition to our findings highlight the importance of local ownership, responsive adaptations and long-term quality control for successful implementation of POCUS.

Regarding equal partnerships, it is noteworthy that in five of the 12 included studies both first and last author originated from a university in a high-income setting [36,37,40,42,44] and only in two studies both were from the country where the research was conducted [41,45]. Although scientific capacity building is not a core component of most implementation frameworks, its inclusion could help to strengthen partnerships and support sustainable research capacity.

Overall, the complexity of conducting implementation programmes in flexible, non-standardised ways while following consistent implementation strategies must not be underestimated, especially given their high demands on required personnel and time.

Nevertheless, WHO has recognised ultrasound as a comparatively affordable imaging modality that strengthens healthcare services [49]. Economic evaluations of established POCUS protocols are therefore particularly important regarding their potential impact in LMICs [50,51]. Such evaluations should extend beyond system-level costs, which dominated the studies in this review, and should include patient access, financial burden, and broader societal benefits [52]. Only one identified study adopted a comprehensive cost-utility analysis incorporating health-system costs and patient-centred outcomes additionally standardising calculations to make them comparable across different settings and over time [44]. The heterogeneity of economic evaluations designs, including differences in input variables, outcome measures, and statistical approaches, made it difficult to draw firm conclusions for policy recommendations.

These findings highlight challenges of evaluating costs of diagnostic interventions as relevant outcome measures were often difficult to define, particularly as they typically inform clinical management indirectly rather than through a direct therapeutic effect [53]. This was further complicated by the settings of the studies included in this review reflected in a variability of clinical pathways and health system capacities.

This review has limitations that are relevant to the interpretation of the findings. First, we were only able to include relatively few studies, many of which were highly context-specific with narrow clinical focus. Study designs were largely cross-sectional, small-scale, and non-randomised, further limiting the strength of the evidence. Additionally, our search strategy was intentionally designed to identify comprehensive studies, rather than those focusing solely on components such as needs assessments, training evaluation, or isolated descriptions of barriers and facilitators. This may have led to an exclusion of studies that reported these aspects separately across several publications. However, for the studies that were included, additional searches were conducted to identify any associated publications that might provide further relevant information and none were found. Second, the CFIR–ERIC matching tool is intended to identify solutions to challenges identified within implementation programmes and our application based on barriers and facilitators reported in scientific publications goes beyond this intended application. It is possible that our included studies did not comprehensively report barriers encountered during implementation and this may have led to biased or incomplete strategy recommendations. However, we attempted to mitigate this bias by additionally including reported facilitators and found a significant overlap in the resulting strategy recommendations. This approach has previously been found to be valid and useful [27,28].

Even in view of these limitations, our review identified several recommendations for future implementation of POCUS in LMICs. Programmes should be co-designed with frontline providers and guided by continuous evaluation and flexible adaptation of programmes to improve sustainability. Inclusion of capacity building for research can ensure continued equitable collaborations and support the expansion of training efforts and equipment access. Future implementation projects for POCUS in LMICs should consider longitudinal evaluation of uptake and impact integrating implementation and economic perspectives to bridge the current disconnect and inform evidence-based decision making.

## Supporting information

**S1 Checklist. PRISMA-ScR checklist.**
(DOCX)

**S1 Text. Search Strategy.**
(DOCX)

**S2 Text. Critical Appraisal Tool.**
(DOCX)

**S3 Text. Quality Assessment.**
(DOCX)

## Author contributions

**Conceptualization:** Grace W. Banda-Katha, Benno Kreuels, Paul Rahden.

**Data curation:** Grace W. Banda-Katha, Timothy Kachitosi, Paul Rahden.

**Formal analysis:** Grace W. Banda-Katha, Paul Rahden.

**Funding acquisition:** Benno Kreuels.

**Methodology:** Grace W. Banda-Katha, Paul Rahden.

**Project administration:** Grace W. Banda-Katha, Benno Kreuels, Paul Rahden.

**Supervision:** Benno Kreuels, Paul Rahden.

**Visualization:** Grace W. Banda-Katha.

**Writing – original draft:** Grace W. Banda-Katha, Paul Rahden.

**Writing – review & editing:** Henry C. Mwandumba, Victor Mwapasa, Lucky G. Ngwira, Benno Kreuels.

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
