## [Decision Letter · Decision Letter 0]

17 Oct 2025

PGPH-D-25-02260

Implementation strategies and Economic Considerations for Point-of-Care Ultrasound in Low-Resource Settings: A Scoping Review

Dear Dr. Kreuels,

Thank you for submitting your manuscript to PLOS Global Public Health. After careful consideration, we feel that it has merit but does not fully meet PLOS Global Public Health’s publication criteria as it currently stands. Therefore, we invite you to submit a revised version of the manuscript that addresses the points raised during the review process.

We look forward to receiving your revised manuscript.

Kind regards,

Asad Naveed

Academic Editor

Journal Requirements:

Additional Editor Comments (if provided):

Reviewers' comments:

Reviewer's Responses to Questions

**Comments to the Author**

1. Does this manuscript meet PLOS Global Public Health’s publication criteria ? Is the manuscript technically sound, and do the data support the conclusions? The manuscript must describe methodologically and ethically rigorous research with conclusions that are appropriately drawn based on the data presented.

Reviewer #1: Partly

Reviewer #2: No

Reviewer #3: Partly

Reviewer #4: Partly

2. Has the statistical analysis been performed appropriately and rigorously?

Reviewer #1: Yes

Reviewer #2: No

Reviewer #3: Yes

Reviewer #4: Yes

3. Have the authors made all data underlying the findings in their manuscript fully available (please refer to the Data Availability Statement at the start of the manuscript PDF file)?

Reviewer #1: Yes

Reviewer #2: Yes

Reviewer #3: Yes

Reviewer #4: Yes

4. Is the manuscript presented in an intelligible fashion and written in standard English?

Reviewer #1: No

Reviewer #2: Yes

Reviewer #3: Yes

Reviewer #4: Yes

5. Review Comments to the Author

Reviewer #1: Regarding the manuscript titled:

Implementation Strategies and Economic Considerations for Point-of-Care Ultrasound in Low-Resource Settings: A Scoping Review.

The authors conducted a scoping review of 12 studies that implemented point-of-care ultrasound (POCUS) in low-resource settings (LRS). Six studies focused on implementation strategies, and six studies evaluated economic aspects. While the topic is important and relevant.

I have the following comments:

Methods:

All the included studies are either cross-sectional or prospective in design; no randomized trials were identified. This should be clearly stated in the limitations.

Introduction:

The opening sentence describing challenges of CT and MRI is redundant and does not add value to the introduction. I recommend removing it. Instead, the introduction should focus more directly on comparing the challenges of comprehensive ultrasound versus POCUS in low-resource settings.

Results:

Please add patient numbers for each study in Tables 1 and 2.

The results section is excessively long (up to 12 pages), which makes it difficult to follow. The authors should prioritize the most relevant findings that directly address the aim of the review. Less essential results could be moved to supplementary material.

Consider avoiding duplication of findings: if results are presented in a table or a figure, there is no need to fully reproduce them in the text.

Discussion:

The discussion is poorly structured. The key findings should be summarized clearly at the beginning, but instead, two pages are spent re-describing results already presented in the Results section.

The authors should review and compare their findings with results from other studies, particularly contrasting experiences in low- and high-income countries.

Formatting:

The manuscript contains inconsistent use of fonts and font sizes. Please ensure a uniform font type and size throughout.

Reviewer #2: The study aims to identify implementation strategies and highlight their challenges and approaches for integration along with the cost effectiveness analysis. The study presented by the authors have considered only 12 studies (6 implementation, 6 economic evaluations) which appears to be inadequate to draw scientific coherent conclusion. The evidence base is found to be thin, fragmented, and dominated by short-term, externally led projects. While the scoping review attempts to address the concerns of POCUS implementation strategies on ground but lacks strong evidence to policy recommendations. The study does not provide definitive guidance on cost-effectiveness or large-scale implementation of POCUS in LMICs.

Only 12 studies were included. Many are small-scale, single-country, short-term interventions with weak designs (mostly observational, no control groups). This limits generalizability. While CFIR-ERIC tool strengthens interpretation, moving beyond description to actionable strategy recommendations, barriers were poorly reported in included studies, forcing the authors to use facilitators “in reverse.” This is innovative but methodologically debatable, as acknowledged by the authors. The studies on implementation strategies are mainly observational; hence, this does not give any recommendations on the implementation strategy to be adopted.

The study has considered limited data on the cost impact analysis from the health system/provider perspective. The study should have included direct & indirect costs involved in POCUS intervention as compared to the existing practice. The study has also not included the incremental cost utility ratio (ICUR) for a better understanding and assessment of the intervention. There is a need to incorporate equity, patient-level costs, and contextual adaptations into economic analyses.

Reviewer #3: Scope and Objectives:

The objectives could be more clearly differentiated between implementation strategy mapping and economic evaluation synthesis. Currently, both are discussed together throughout the text, which can make it difficult to follow the distinct analytical processes.

Consider briefly stating in the Methods how these two review components were conceptually linked.

Methodological Transparency:

The search end date is listed as April 30, 2025—please clarify if this is a projected cut-off or a typographical error.

Provide the full search strategy as a supplementary file (as referenced in OSF) to enhance reproducibility.

Application of the CFIR–ERIC Tool:

The adaptation of the tool to infer facilitators is innovative but unconventional. Please elaborate on how this reverse application was validated or justified, with references to prior work.

Results:

The quality assessment results could be summarised in one paragraph rather than dispersed across sections.

Discussion:

The discussion could better integrate how identified implementation strategies align with or diverge from prior POCUS training reviews (e.g., Eppel et al., 2025).

Minor Comments

1. Proofread for consistency in referencing style and punctuation (some references lack DOIs or journal issue numbers).

2. Standardize terminology (e.g., “low-resource settings,” “LMICs,” “resource-constrained settings”).

3. The abstract could be more quantitative—e.g., report the number of included studies per region and methodological quality category.

4. Ensure figure legends are self-contained and that figure quality meets journal standards.

5. Minor English language polishing would improve readability, particularly in the Results and Discussion sections.

Reviewer #4: MINOR REVISION:

- Authors should specify the year in which the World Bank classified these countries as LMICs, given that this is dynamic.

- Explain the cleaning procedure in Excel (line 157).

- Explain how the checklists were adapted (line 178). This will make it easier to understand.

- The results are essentially descriptive; consider linking the Implementation and Economy dimensions.

- Give more weight to your approach on the reverse use of CFIR-ERIC.

- Discuss the limitations of the CFIR-ERIC tool.

MAJOR REVISION:

- To ensure the quality of the study selection process, inter-rater reliability should have been determined;

- Line 180: is the scoring standard? If not, explain the reasons for this adaptation and how it was done.

6. PLOS authors have the option to publish the peer review history of their article (what does this mean? ). If published, this will include your full peer review and any attached files.

**Do you want your identity to be public for this peer review?** For information about this choice, including consent withdrawal, please see our Privacy Policy .

Reviewer #1: No

Reviewer #2: No

Reviewer #3: No

Reviewer #4: No

Figure Resubmissions:

---

## [Decision Letter · Decision Letter 1]

16 Dec 2025

PGPH-D-25-02260R1

Implementation strategies and Economic Considerations for Point-of-Care Ultrasound in Low-and-Middle-Income Countries: A Scoping Review

Dear Dr. Kreuels,

Thank you for submitting your manuscript to PLOS Global Public Health. After careful consideration, we feel that it has merit but does not fully meet PLOS Global Public Health’s publication criteria as it currently stands. Therefore, we invite you to submit a revised version of the manuscript that addresses the points raised during the review process.

The manuscript has been evaluated by three reviewers, and their comments are available below. The reviewers have raised a number of remaining concerns that need attention in relation to the language used within the manuscript.

Could you please revise the manuscript to carefully address the concerns raised?

We look forward to receiving your revised manuscript.

Kind regards,

Jen Edwards

Staff Editor

Journal Requirements:

1. “PRISMA-ScR checklist-291125.docx and “Raw_Data Extraction.xlsx” are currently uploaded as an 'Other' file type, which is not viewable by reviewers. Please ensure that all files meant for review are uploaded as 'Supporting Information' and include a legend in the manuscript.

2. We have noticed that you have uploaded Supporting Information files, but you have not included a list of legends. Please add a full list of legends for your Supporting Information files before or after the references list.

3. Please provide separate main figure files in .tif or .eps format only and remove any figures embedded in your manuscript file. Please also ensure that all files are under our size limit of 10MB. Please leave the figure captions in the manuscript.

4. Some material included in your submission may be copyrighted. According to PLOS’s copyright policy, authors who use figures or other material (e.g., graphics, clipart, maps) from another author or copyright holder must demonstrate or obtain permission to publish this material under the Creative Commons Attribution 4.0 International (CC BY 4.0) License used by PLOS journals. Please closely review the details of PLOS’s copyright requirements here: PLOS Licenses and Copyright. If you need to request permissions from a copyright holder, you may use PLOS's Copyright Content Permission form.

Potential Copyright Issues:

Figure 2: please (a) provide a direct link to the base layer of the map (i.e., the country or region border shape) and ensure this is also included in the figure legend; and (b) provide a link to the terms of use / license information for the base layer image or shapefile. We cannot publish proprietary or copyrighted maps (e.g. Google Maps, Mapquest) and the terms of use for your map base layer must be compatible with our CC-BY 4.0 license.

Additional Editor Comments (if provided):

Reviewers' comments:

Reviewer's Responses to Questions

**Comments to the Author**

1. If the authors have adequately addressed your comments raised in a previous round of review and you feel that this manuscript is now acceptable for publication, you may indicate that here to bypass the “Comments to the Author” section, enter your conflict of interest statement in the “Confidential to Editor” section, and submit your "Accept" recommendation.

Reviewer #2: All comments have been addressed

Reviewer #3: All comments have been addressed

Reviewer #4: All comments have been addressed

2. Does this manuscript meet PLOS Global Public Health’s publication criteria ? Is the manuscript technically sound, and do the data support the conclusions? The manuscript must describe methodologically and ethically rigorous research with conclusions that are appropriately drawn based on the data presented.

Reviewer #2: Yes

Reviewer #3: Yes

Reviewer #4: Yes

3. Has the statistical analysis been performed appropriately and rigorously?

Reviewer #2: Yes

Reviewer #3: Yes

Reviewer #4: Yes

4. Have the authors made all data underlying the findings in their manuscript fully available (please refer to the Data Availability Statement at the start of the manuscript PDF file)?

Reviewer #2: Yes

Reviewer #3: Yes

Reviewer #4: Yes

5. Is the manuscript presented in an intelligible fashion and written in standard English?

Reviewer #2: Yes

Reviewer #3: Yes

Reviewer #4: Yes

6. Review Comments to the Author

Reviewer #2: The authors may like to reframe the title by excluding 'economic considerations' as the study does not include CEA or ICER/ICUR findings/result.

Reviewer #3: please review the grammar mistakes:

“low-and-middle-income countries” → “low- and middle-income countries”

(Lines 69–70 and elsewhere)

“high quality” → “high-quality” (line 75)

Line 71–72

“Given its comparatively low financial and technical requirements(,)”

(add comma)

Lines 87–90

“driven by different stakeholders with diverse aims and priorities of individual programmes”

Grammatically awkward. Suggested rewrite:

“driven by different stakeholders with diverse aims and priorities across individual programmes

Line 102–103

“CFIR - Expert Recommendations for Implementing Change (CFIR103 ERIC)”

This appears to be a line-break error.

correct: “CFIR–Expert Recommendations for Implementing Change (CFIR–ERIC)

Lines 121–122

“as well as economic evaluations of such as these components inform the feasibility”

This sentence is grammatically incorrect.

Line 123–125

“published before 15th of February 2024”

Formatting error.

correct: “published before 15 February 2024”

Line 252

“One study reported number of scans”

Correct: “One study reported the number of scans”

“patients centered outcomes” (line 408) Correct to:

“patient-centred outcomes” (UK spelling, consistent with “programme”)

Table 1

“Crosssectional” → “Cross-sectional” (multiple instances)

Impression:

No major mistakes.

The manuscript mainly needs:

Correction of a few grammar and formatting errors

Hyphenation consistency

Minor clarity and style refinements

Reviewer #4: (No Response)

7. PLOS authors have the option to publish the peer review history of their article (what does this mean? ). If published, this will include your full peer review and any attached files.

**Do you want your identity to be public for this peer review?** For information about this choice, including consent withdrawal, please see our Privacy Policy .

Reviewer #2: **Yes:** Dr Ranjan Choudhury

Reviewer #3: No

Reviewer #4: No

 Figure Resubmissions:

---

## [Editor Report · Decision Letter 2]

5 Jan 2026

Implementation strategies and Economic Considerations for Point-of-Care Ultrasound in Low- and Middle-Income Countries: A Scoping Review

PGPH-D-25-02260R2

Dear Dr Kreuels,

We are pleased to inform you that your manuscript 'Implementation strategies and Economic Considerations for Point-of-Care Ultrasound in Low- and Middle-Income Countries: A Scoping Review' has been provisionally accepted for publication in PLOS Global Public Health.

Best regards,

Julia Robinson

Executive Editor